# Genetic risk for neurodegenerative conditions is linked to disease-specific microglial pathways

**Aydan Askarova**[1,2], **Reuben M. Yaa**[1,2], **Sarah J. Marzi**[3,4], **Alexi Nott**[1,2]*

1 Department of Brain Sciences, Imperial College London, London, United Kingdom, 2 United Kingdom Dementia Research Institute, Imperial College London, London, United Kingdom, 3 Department of Basic and Clinical Neuroscience, Institute of Psychiatry, Psychology and Neuroscience, King's College London, London, United Kingdom, 4 United Kingdom Dementia Research Institute, King's College London, London, United Kingdom

* a.nott@imperial.ac.uk

## Abstract

Genome-wide association studies have identified thousands of common variants associated with an increased risk of neurodegenerative disorders. However, the noncoding localization of these variants has made the assignment of target genes for brain cell types challenging. Genomic approaches that infer chromosomal 3D architecture can link noncoding risk variants and distal gene regulatory elements such as enhancers to gene promoters. By using enhancer-to-promoter interactome maps for human microglia, neurons, and oligodendrocytes, we identified cell-type-specific enrichment of genetic heritability for brain disorders through stratified linkage disequilibrium score regression. Our analysis suggests that genetic heritability for multiple neurodegenerative disorders is enriched at microglial chromatin contact sites, while schizophrenia heritability is predominantly enriched at chromatin contact sites in neurons followed by oligodendrocytes. Through Hi-C coupled multimarker analysis of genomic annotation (H-MAGMA), we identified disease risk genes for Alzheimer's disease, Parkinson's disease, multiple sclerosis, amyotrophic lateral sclerosis and schizophrenia. We found that disease-risk genes were overrepresented in microglia compared to other brain cell types across neurodegenerative conditions and within neurons for schizophrenia. Notably, the microglial risk genes and pathways identified were largely specific to each disease. Our findings reinforce microglia as an important, genetically informed cell type for therapeutic interventions in neurodegenerative conditions and highlight potentially targetable disease-relevant pathways.

## Author summary

Neurodegenerative diseases, including Alzheimer's disease, Parkinson's disease, multiple sclerosis, and amyotrophic lateral sclerosis, are complex conditions influenced by genetic factors. While previous studies have identified many genetic variants linked to these disorders, most are located in noncoding regions of the genome, making it difficult to determine which genes they affect. To address this, we used a 3D genomic approach

**Data availability statement:** Code is available: https://github.com/aydanasg/cell_hmagma. PLAC-seq, H3K27ac ChIP-seq, H3K4me3 ChIP-seq and ATAC-seq datasets were taken from (1) and processed data is available: https://github.com/nottalexi/brain-cell-type-peak-files.

**Funding:** AN and SJM are supported by the UK Dementia Research Institute [award number UKDRI- 5208 to AN; UKDRI-6009 to SJM] through UK DRI Ltd, principally funded by the Medical Research Council. AN, SJM and RMY received salaries from the UK Dementia Research Institute. AN and SJM are supported by the Edmond and Lily Safra Early Career Fellowship Program. AN is supported by The Dunhill Medical Trust [grant number AISRPG2305\26]. AA received a stipend funded by the Imperial College London President's PhD Scholarships. The funders had no role in study design, data collection and analysis, decision to publish, or preparation of the manuscript.

**Competing interests:** The authors have declared that no competing interests exist.

to map how these genetic variants interact with genes in different brain cell types. Our analysis revealed that genetic risk for Alzheimer's disease, Parkinson's disease, multiple sclerosis, and amyotrophic lateral sclerosis is particularly enriched in microglia, the brain's immune cells, while schizophrenia risk is more strongly linked to neurons and oligodendrocytes. By integrating our findings with chromatin looping data, we identified key genes that may contribute to disease development. Interestingly, microglial risk genes varied between diseases, suggesting that each condition involves distinct biological pathways. Our work reinforces the idea that microglia play a central role in neurodegenerative diseases and highlights potential new targets for treatments.

## Introduction

Genetics plays a significant role in the etiology of neurodegenerative disorders including Alzheimer's disease (AD), Parkinson's disease (PD), multiple sclerosis (MS) and amyotrophic lateral sclerosis (ALS) [1–10]. Familial forms have been identified for AD, PD and ALS that exhibit Mendelian patterns of inheritance and are associated with rare variants with strong effect sizes [11–26]. While the genetics underlying familial cases have been informative in our understanding of disease etiology, most individuals presenting with neurogenerative disorders, including MS, have sporadic forms of the disease. Genome-wide association studies (GWAS) for sporadic neurodegenerative disorders have identified thousands of common variants associated with an increased risk of disease and highlight the heterogeneity of these disorders [27–33]. GWAS risk variants generally have a relatively high prevalence in the population and exhibit smaller effect sizes, with their risk contribution believed to arise from the combined effects of multiple variants [34].

Most GWAS risk variants reside within non-coding regions of the genome and are often located distally from the nearest known genes [35]. GWAS risk variants are enriched at chromatin accessibility regions that likely function as gene regulatory elements such as enhancers and promoters [36,37]. Enhancers are distal genomic regions associated with chromatin accessibility and are characterized by the presence of specific histone modifications, including acetylation of histone H3 lysine 27 (H3K27ac) [38]. Enhancers are highly cell type-specific [39] and can be incorporated into heritability analysis to prioritize cell types associated with the genetic risk of complex traits. GWAS variants for neurological disorders and psychiatric traits have been associated with cell type-specific heritability enrichment. For example, AD risk variants were found to be enriched in microglia and macrophage enhancers, while schizophrenia risk was enriched in neuronal gene regulatory regions [40–47].

Enhancers have been informative for the allocation of cell types associated with genetic risk. However, the distal localization of GWAS risk variants has made the identification of target genes impacted by these variants a major challenge. The mammalian genome has a non-random three-dimensional organization that connects distal chromosomal regions through the formation of chromatin loops [48]. Functional chromatin interactions include the association of gene promoters with *cis*-regulatory regions, such as enhancers [48]. The recruitment of transcription factors and structural proteins to enhancers and their interaction with promoters facilitates the formation of the pre-initiation complex and gene transcription [49,50]. Genetic variants localized to gene regulatory regions were thought to disrupt enhancer function or enhancer-to-promoter interactions, ultimately impacting gene expression and cell behavior [46,51]. Similar to enhancers, chromatin interactions are cell-type-specific [40]. Hence, localization of non-coding GWAS variants to chromatin contact sites

could predict cell type-specific genes and pathways that are susceptible to genetic variation in neurodegenerative disorders.

Enhancer-to-promoter interactomes are available for three of the major brain cell types, however, the assignment of GWAS risk variants to genes has been hindered by a lack of computational tools. A recently developed tool, Hi-C coupled multimarker analysis of genomic annotation (H-MAGMA), identifies putative disease risk genes by accounting for GWAS variants within distal non-coding regions [52]. H-MAGMA predicts gene-level associations with diseases by combining GWAS summary statistics with enhancer-to-promoter interactomes [52,53]. Here we used H-MAGMA coupled with chromatin data to map out disease genes for neurodegenerative diseases. By integrating epigenetic annotations with chromatin interaction data, we identified putative cell types and genes that contribute to the genetic susceptibility of these disorders. We found that risk genes are enriched in microglia across multiple neurodegenerative diseases (AD, PD, ALS, and MS). However, the pathways impacted by microglial H-MAGMA-risk genes are mostly unique for each disorder, indicating that immune processes exhibit disease-specific patterns.

## Results

### Microglial chromatin interactions are enriched for genetic risk variants associated with neurodegenerative disorders

To determine whether disease risk variants for neurodegenerative disorders are associated with genes linked to distal gene regulatory regions, we used proximity ligation-assisted chromatin immunoprecipitation-seq (PLAC-seq) data generated from human cortical neurons, microglia and oligodendrocytes [40]. Histone H3 lysine 27 acetylation (H3K27ac) is a histone modification enriched at active gene promoters and distal enhancers, whereas histone H3 lysine 4 trimethylation (H3K4me3) is enriched at promoters only [54,55]. By integrating H3K4me3-anchored PLAC-seq-defined chromatin loops with ATAC-seq, H3K27ac chromatin immunoprecipitation (ChIP)-seq and H3K4me3 ChIP-seq from the same cell types [40], we classified chromatin interactions as either: i) promoter-to-enhancer; ii) promoter-to-promoter; iii) promoter-to-ATAC; iv) promoter-to-promoter/enhancer; v) promoter-to-other; vi) H3K4me3-to-H3K4me3; vii) H3K4me3-to-other; and viii) other interactions. Promoter-to-enhancer loops were the most common classification of chromatin interactions for each cell type, representing 29.4%, 39.1% and 38.2% of interactions in microglia, neurons, and oligodendrocytes, respectively (Fig 1a). The next most abundant classifications were chromatin interactions that occurred at promoters-to-other or H3K4me3-to-other (Fig 1a). Promoters are known to interact with more than one enhancer. For microglia, neurons and oligodendrocytes, most promoters interacted with more than one enhancer and for enhancer-to-promoter interactions, most enhancers interacted with a single promoter (Fig 1b). The average distance of these enhancer-to-promoter interactions was 200 kb for microglia (range: 3,443 – 1,003,377 bp; mean: 206,304 bp), 250 kb for neurons (2,657 – 1,008,329 bp; mean: 254,016 bp) and 150 kb for oligodendrocytes (range: 2,974 – 999,211 bp; mean: 159,230 bp) (Fig 1c). Overall, H3K4me3-anchored PLAC-seq chromatin loops in microglia, neurons and oligodendrocytes predominantly identified promoters that were linked to multiple distal enhancers.

To examine whether disease heritability was enriched at brain cell type chromatin interactions, we used stratified linkage disequilibrium score (sLDSC) regression analysis. Cell type disease enrichment by sLDSC regression was assessed using chromatin interactions defined as (i) all PLAC-seq bins irrespective of functional genomic annotations (total PLAC-seq bins), (ii) PLAC-seq bins subset to both active gene promoters (H3K4me3 + H3K27ac) and distal enhancers (H3K27ac only) (promoter & enhancer PLAC-seq bins), (iii) PLAC-seq bins

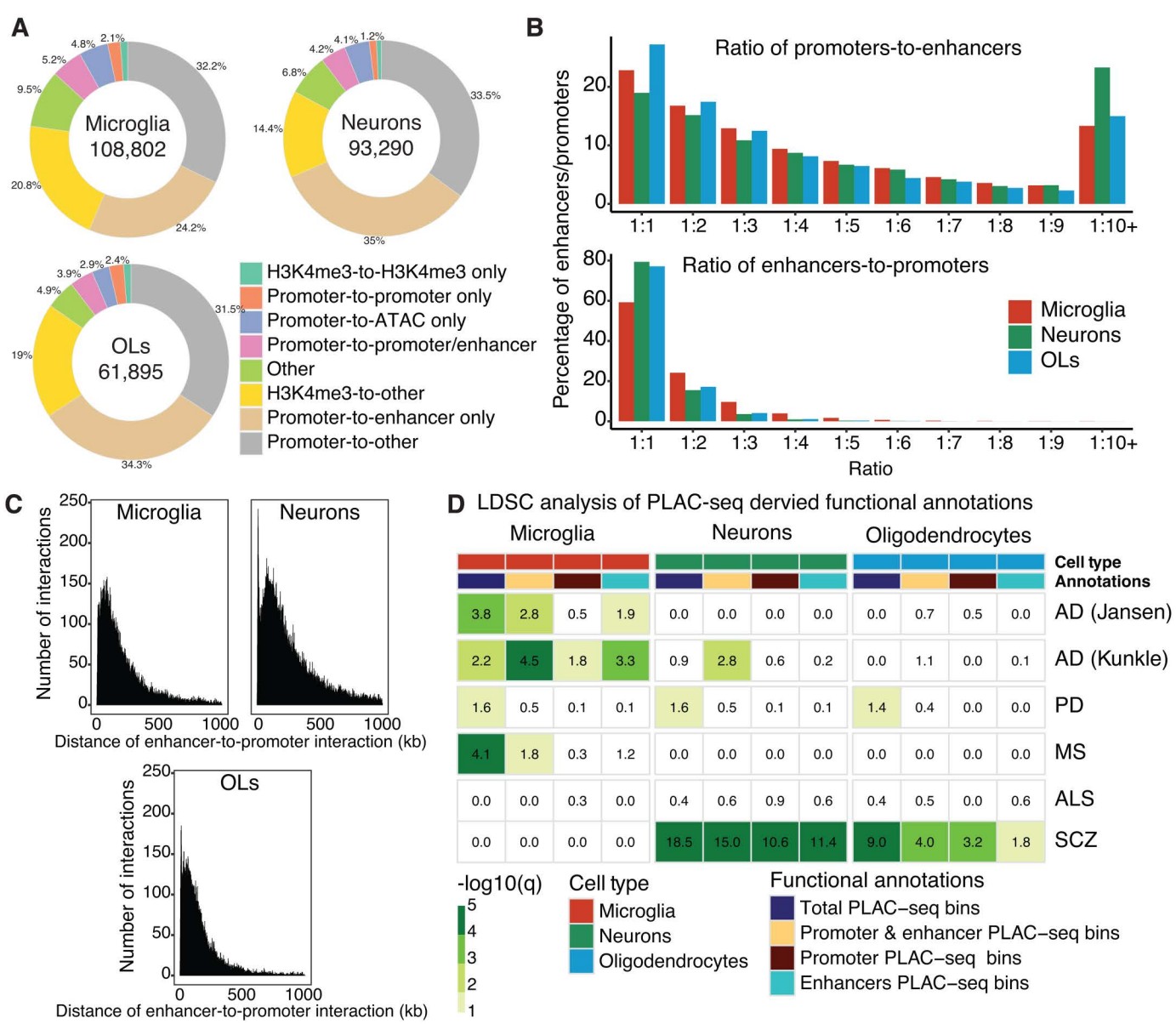

**Fig 1. Microglia enhancer-to-promoter interactions were enriched for disease-risk variants across multiple neurodegenerative conditions.** A) Doughnut plots of classifications of PLAC-seq interactions identified in human microglia, neurons and oligodendrocytes [40] with the total number of interactions shown in the center. 'Promoters', PLAC-seq bins that overlap a H3K4me3 and H3K27ac peak within 2,000 bp of a transcriptional start site (TSS). 'Enhancers', PLAC-seq bins that overlap H3K27ac peaks distal to the TSS. 'H3K4me3', PLAC-seq bins that overlap H3K4me3 peaks distal to TSS. 'ATAC', PLAC-seq bins that overlap chromatin accessible regions devoid of H3K4me3 and H3K27ac. B) Percent distribution of the number of enhancers interacting with individual promoters (top plot) and the number of promoters interacting with individual enhancers (bottom plot). C) Distribution plot of the proportion of distances between midpoints of promoters and midpoints of enhancers that interact based on chromatin interaction PLAC-seq data. D) Heatmap of partitioned heritability sLDSC regression analysis of PLAC-seq-derived functional annotations: (i) total PLAC-seq bins, (ii) promoter & enhancer PLAC-seq bins, (iii) promoter PLAC-seq bins and (iv) enhancer PLAC-seq bins for microglia, neurons and oligodendrocytes in AD [29,30], PD [27] (excluding 23andMe), MS [33], ALS [31], and schizophrenia [32]. Shown are LDSC enrichment p-values with Benjamini–Hochberg FDR correction for the number of diseases and cell types (-log10(q)). Disease enrichment was considered insignificant if the coefficient z-score was negative and assigned a 0.0 -log10(p) score. OLs, oligodendrocytes; SCZ, schizophrenia.

subset to active gene promoters (promoter PLAC-seq bins) and (iv) PLAC-seq bins subset to distal enhancers (enhancer PLAC-seq bins). Cell type disease enrichment was assessed using summary statistics from two complementary AD GWAS; one study was based exclusively

on clinical diagnosis [30], while the second included by-proxy cases [29]. GWAS summary statistics were analyzed for three additional neurodegenerative conditions, PD, MS and ALS [27,31,33] and for a neurodevelopmental condition, schizophrenia [32].

Microglia PLAC-seq bins showed enrichment for AD risk variants over other cell types, with a greater enrichment of AD risk variants found at enhancer PLAC-seq bins compared to promoter PLAC-seq bins (Figs 1d and S1). These findings corroborate observations that AD GWAS variants are enriched at microglia enhancers compared to microglia promoters defined using histone modifications [40,56]. The microglia enhancer PLAC-seq bins are likely physically linked to gene promoters and therefore functionally relevant. An enrichment of disease risk variants at microglia PLAC-seq bins was also observed for PD and MS, although there was no clear preference for either promoter or enhancer interacting regions for these disorders (Figs 1d and S1). No enrichment for ALS risk variants was identified at PLAC-seq bins for microglia, neurons or oligodendrocytes (Figs 1d and S1). In contrast, for schizophrenia, heritability showed a strong enrichment of disease risk at PLAC-seq bins identified in neurons and oligodendrocytes (Figs 1d and S1). The heritability enrichment for schizophrenia was stronger in neurons than oligodendrocytes (promoter & enhancer PLAC-seq bins; sLDSC; neurons -log10(q)=15; oligodendrocytes, -log10(q)=4.0) (Fig 1d). This supports previous findings showing that schizophrenia GWAS variants were enriched at neuronal promoters and enhancers using annotations defined by histone modifications [40,44]. Overall, chromatin-interacting regions in microglia show a broad enrichment for disease heritability across multiple neurodegenerative disorders.

## Microglial chromatin interactions identify disease risk genes across multiple neurodegenerative conditions

Promoter-to-enhancer interactions link distal gene regulatory regions, such as enhancers, to active gene promoters and can be used to infer disease-risk genes for noncoding GWAS risk variants. H-MAGMA was used to identify disease-risk genes in microglia, neurons and oligodendrocytes for AD, PD, MS, ALS and schizophrenia by incorporating PLAC-seq interactomes for the corresponding cell types (S2 Fig). In all the neurodegenerative GWAS that we assessed, the highest number of risk genes were identified in microglia compared to neurons and oligodendrocytes (Fig 2a and S1 Table). In contrast, for schizophrenia, the highest number of risk genes were identified in neurons (Fig 2a and S1 Table). Schizophrenia had the highest number of H-MAGMA risk genes across all cell types compared to neurodegenerative disorders. The number of identified risk genes is partially influenced by the number of associated risk variants reported in the corresponding GWAS. Schizophrenia had the most variants with a GWAS p-value below $5 \times 10^{-8}$ (19,898 SNPs), more than five times the number reported for the second-highest GWAS analysed in this study (3,555 SNPs in MS).

The number of PLAC-seq chromatin interactions were higher in microglia compared to other cell types (microglia, 108802; neurons, 93290; oligodendrocytes, 61895; Fig 1a), which may partially explain the increased number of microglia disease risk genes identified across neurodegenerative conditions. To account for the differing number of chromatin interactions identified between the three cell types, the PLAC-seq data was randomly downsampled to 60,000 chromatin interactions per cell type. This was followed by H-MAGMA analysis, which was repeated for 10 iterations (Fig 2b). H-MAGMA analysis using the 60,000 downsampled PLAC-seq chromatin interactions maintained a similar distribution of disease-risk genes across the three cell types (Fig 2b). Importantly, when the number of chromatin interactions was the same for each cell type, the number of disease-risk genes identified remained highest in microglia for AD, PD, MS and ALS (Fig 2b).

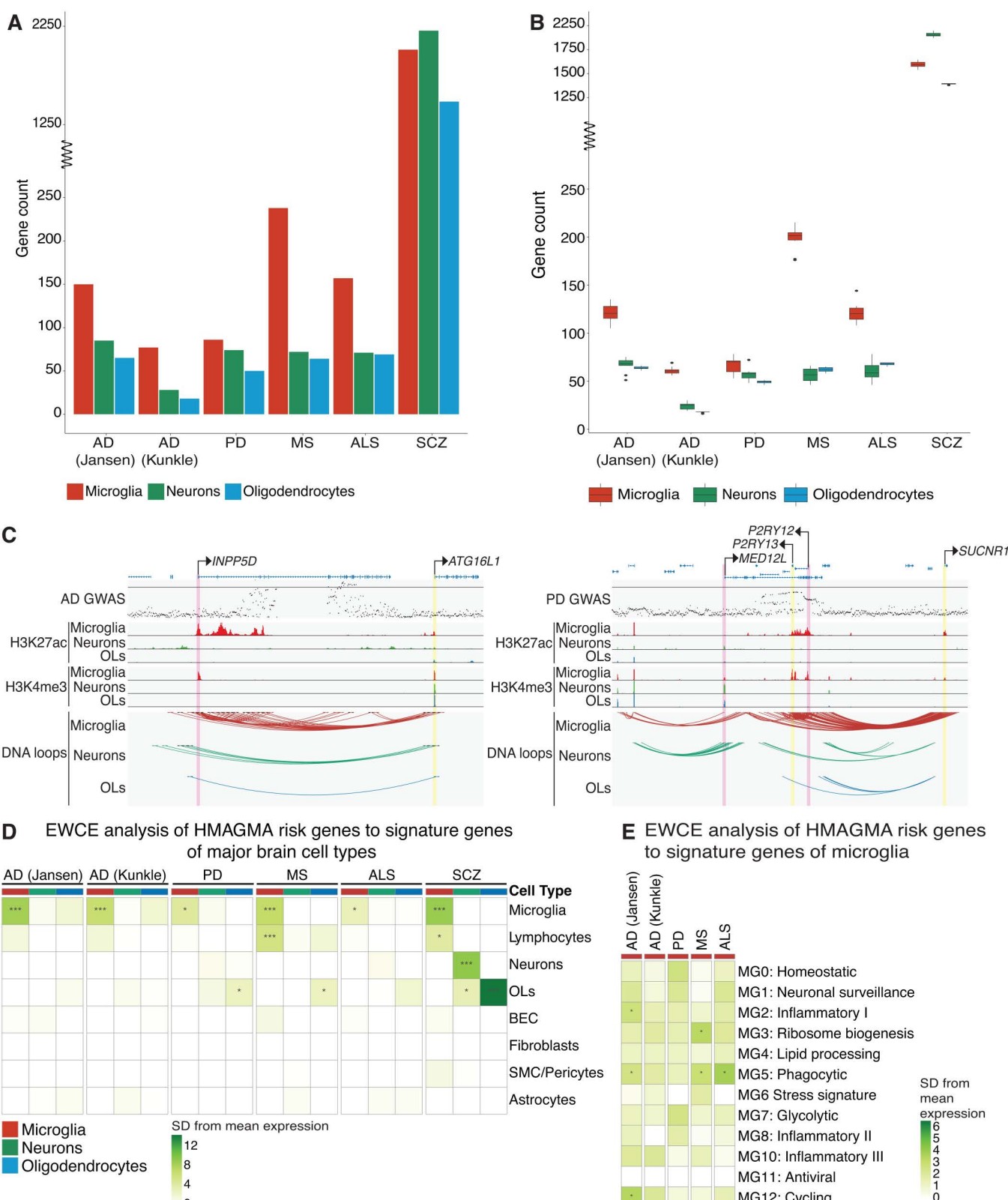

**Fig 2. Microglial disease risk genes were identified for distal GWAS variants using chromatin loops.** A) The number of disease risk genes identified in microglia, neurons and oligodendrocytes using H-MAGMA and GWAS for AD, PD (excluding 23andMe), MS, ALS, and schizophrenia. Gene-to-SNP associations were assigned for SNPs that were located within the promoter or exon of a gene, and within enhancers that were linked to genes through PLAC-seq interactions. B) The number of disease risk genes identified using the sampled down loops for microglia, neurons and oligodendrocytes with H-MAGMA for

AD, PD (excluding 23andMe), MS, ALS, and schizophrenia. To account for differences in chromatin interactions between cell types, the number of enhancer-to-promoter interactions was randomly sampled down to 60,000 loops 10 times. Dunn's test (non-parametric) between cell types within each group: AD (Jansen 2019): microglia-neurons (**), microglia-oligodendrocytes (****), neurons-oligodendrocytes (ns); AD (Kunkle 2019): microglia-neurons (*), microglia-oligodendrocytes (****), neurons-oligodendrocytes (*); PD: microglia-neurons (ns), microglia-oligodendrocytes (*****), neurons-oligodendrocytes (**); MS: microglia-neurons (****), microglia-oligodendrocytes (**), neurons-oligodendrocytes (ns); ALS: microglia-neurons (****), microglia-oligodendrocytes (**), neurons-oligodendrocytes (ns); schizophrenia: microglia-neurons (*), microglia-oligodendrocytes (*), neurons-oligodendrocytes (****). C) UCSC browser visualization of the AD *INPP5D* locus and the PD *MED12L* locus. Top track, GWAS SNPs (line indicates GWAS p-value −log10(5e-8) =7.301); middle tracks H3K27ac and H3K4me3 ChIP-seq; bottom track, PLAC-seq interactomes [40]. Yellow, promoters of risk genes identified only by H-MAGMA; pink, promoters of risk genes identified by H-MAGMA and MAGMA. D) EWCE analysis identified cell type enrichment of H-MAGMA disease risk genes from Fig 2A using single-cell RNA-seq from human brain cortical tissue from Tsartsalis et al. (2024) [58]. E) EWCE analysis identified microglia subtype enrichment of microglia H-MAGMA disease risk genes using single-cell RNA-seq of human microglia subtypes from Sun et al. (2023) [59]. EWCE heatmap colour scale represents the number of standard deviations from the mean level of expression found in the target gene lists relative to the bootstrapped mean. SCZ, schizophrenia; OLs, oligodendrocytes.; EWCE p-values *p <0.05, **p<5e-8, ***p=0.

Standard gene mapping approaches such as MAGMA rely on the proximity of GWAS risk variants to genes [53]. The number of genes identified by assigning variants located within 10 kb of a gene using MAGMA was 2–3-fold higher than the number of H-MAGMA microglia risk genes identified across the neurodegenerative conditions (S3 Fig and S3 Table). The higher number of MAGMA risk genes likely reflects the lack of cell type specificity that is provided by incorporating microglia enhancer-to-promoter interactomes. However, many H-MAGMA risk genes were not identified by MAGMA (AD Jansen, 62; AD Kunkle, 32; PD, 30; MS, 75; ALS 68 genes) (S3 Fig and S4 and S5 Tables), including the autophagy-related gene *ATG16L1* at the *INPP5D* AD locus and purinergic receptor gene *P2RY13* located at the PD *MED12L* locus (Fig 2c). These more distal H-MAGMA risk genes were not identified using conventional proximity-based strategies for annotating variants to genes. In summary, H-MAGMA disease risk genes were overrepresented in microglia for neurodegenerative disorders compared to neurons for schizophrenia, which is consistent with the cell type distribution of disease heritability identified using sLDSC regression analysis (Fig 1d).

GWAS risk variants may be differentially enriched at chromatin interaction contact sites at enhancers (PLAC-seq bins at intergenic and intronic regions) compared to promoters (PLAC-seq bins at promoters and exonic regions). To determine GWAS risk enrichment across these gene regulatory classifications, H-MAGMA was repeated using PLAC-seq bins subset to enhancers or promoters. For AD, PD and MS, the number of disease-risk genes identified using microglial enhancer contact sites (100, 64, 179 genes respectively) was comparable to the number of disease-risk genes identified using microglial promoter contact sites (111, 62, 175 genes respectively) (S4 Fig). In ALS, more disease-risk genes were identified using microglia promoter contact sites (124 genes) compared to enhancer contact sites (47 genes) (S4 Fig). This suggests that promoters may play a more crucial role in the genetic risk associated with ALS, in contrast to the significance of enhancers for GWAS risk in other neurodegenerative conditions. Lastly, for schizophrenia, the highest number of disease-risk genes were identified at neuronal promoter contact sites compared to enhancers (S4 Fig).

Disease-risk variants often colocalise with gene regulatory regions that are highly cell-type specific, thereby conferring cell-type-associated genetic susceptibility [36,57]. However, the downstream genes associated with these regulatory regions may be expressed exclusively in the disease-associated cell type or across multiple cell types. Expression Weighted Celltype Enrichment (EWCE) analysis was used to determine the cell type expression of the GWAS risk genes identified by H-MAGMA by incorporating single-cell gene expression data from human cortical brain tissue [58]. EWCE analysis revealed that the expression of microglia H-MAGMA-risk genes for all diseases was enriched predominantly in microglia, with some enrichment in lymphocytes for MS and schizophrenia (Fig 2d). In contrast, H-MAGMA-risk

genes identified in neurons and oligodendrocytes across the neurodegenerative conditions generally did not exhibit a cell type enrichment in gene expression. Only oligodendrocyte H-MAGMA risk genes identified for PD and MS showed an enrichment in oligodendrocyte signature genes (Fig 2d). Disease risk genes identified by H-MAGMA across all three cell types for schizophrenia were characterized by matching cell type-specific gene expression (Fig 2d). Human microglia exhibit heterogeneous transcriptional profiles, a subset of which have been associated with disease conditions [59]. EWCE analysis indicated that microglial AD H-MAGMA risk genes were associated with inflammatory, phagocytic and cycling transcriptional profiles (Fig 2e). In contrast, MS microglial H-MAGMA risk genes were associated with ribosome biogenesis and phagocytic transcriptional profiles and ALS with phagocytic only (Fig 2e).

## Microglial genetic-susceptibility genes are associated with disease-specific pathways

Genetic heritability estimates using sLDSC and the identification of putative GWAS risk genes identified using H-MAGMA highlight the importance of microglia in the genetic susceptibility of neurodegenerative conditions. This may suggest shared dysregulated microglial processes across these disorders. However, an intersection of GWAS-risk genes identified using H-MAGMA for microglia showed a minimal overlap between the different diseases (Fig 3a), including for a subset of the top significant H-MAGMA risk genes (filtered on H-MAGMA p-value; AD, PD, MS, ALS p<5e-8 and schizophrenia p<5e-12) (Fig 3b). Similarly, there was a minimal overlap across diseases for GWAS risk genes identified for neurons and oligodendrocytes (Fig 3a and 3b). While most risk genes were unique to each disorder, some genes were shared across two or more conditions. For example, the major histocompatibility complex (MHC) was identified as a disease-risk locus in MS and schizophrenia (Fig 3b). Disease-risk genes that overlapped across PD, ALS and schizophrenia were *KANSL1-AS1* (microglia and oligodendrocytes) and *KANSL1*, *ARHGAP27*, and *PLEKHM1* (microglia). Interestingly, *KANSL1* and *ARHGAP27* were identified as comorbid genes for PD and ALS [60]. The microglial H-MAGMA-risk genes *BAG6*, *NEU1*, *PRRC2A*, *PSMB8*, *PSMB8-AS1* and *PSMB9* were associated with MS, ALS and schizophrenia. *PSMB8-AS1* was also identified as a microglial risk gene for AD. These findings indicate that microglia are an important cell type associated with genetic susceptibility across multiple neurodegenerative disorders. However, the microglial genes that are impacted by genetic risk are mostly disease specific.

We next assessed specific cellular and biological pathways associated with microglia H-MAGMA-risk genes for each disorder using gene ontology (GO) analysis. GO pathways linked to H-MAGMA-risk genes were mostly unique for each neurodegenerative condition (Fig 3c). This is consistent with the observation that most disease-risk genes were unique to each GWAS (Fig 3a and 3b). The top GO pathways associated with microglial AD-risk genes included lipoproteins, amyloid processing and endocytosis (Fig 3c and S2 Table) compared to neuronal and oligodendrocyte AD-risk genes which were associated with amyloid and tau protein catabolic processes (S5 and S6 Figs). PD microglial-risk genes were associated with the endolysosomal/autolysosomal pathways, synaptic vesicles and epigenetic signaling (Figs 3c and S2). Whereas substantia nigra gliosis, epigenetic signaling and synaptic vesicle pathways were evident in neuronal PD-risk genes, reinforcing the vulnerability of the midbrain in PD (S5 Fig). Both microglia and oligodendrocyte MS risk-genes were associated with MHC protein complexes, autoimmunity, and antigen presentation and processing (Figs 3c and S6 and S2 Table). Risk genes assigned to the MHC Class II complex were also associated with AD and PD, as well as MS (Fig 3b). ALS exhibited associations with vacuoles and kinases, while also

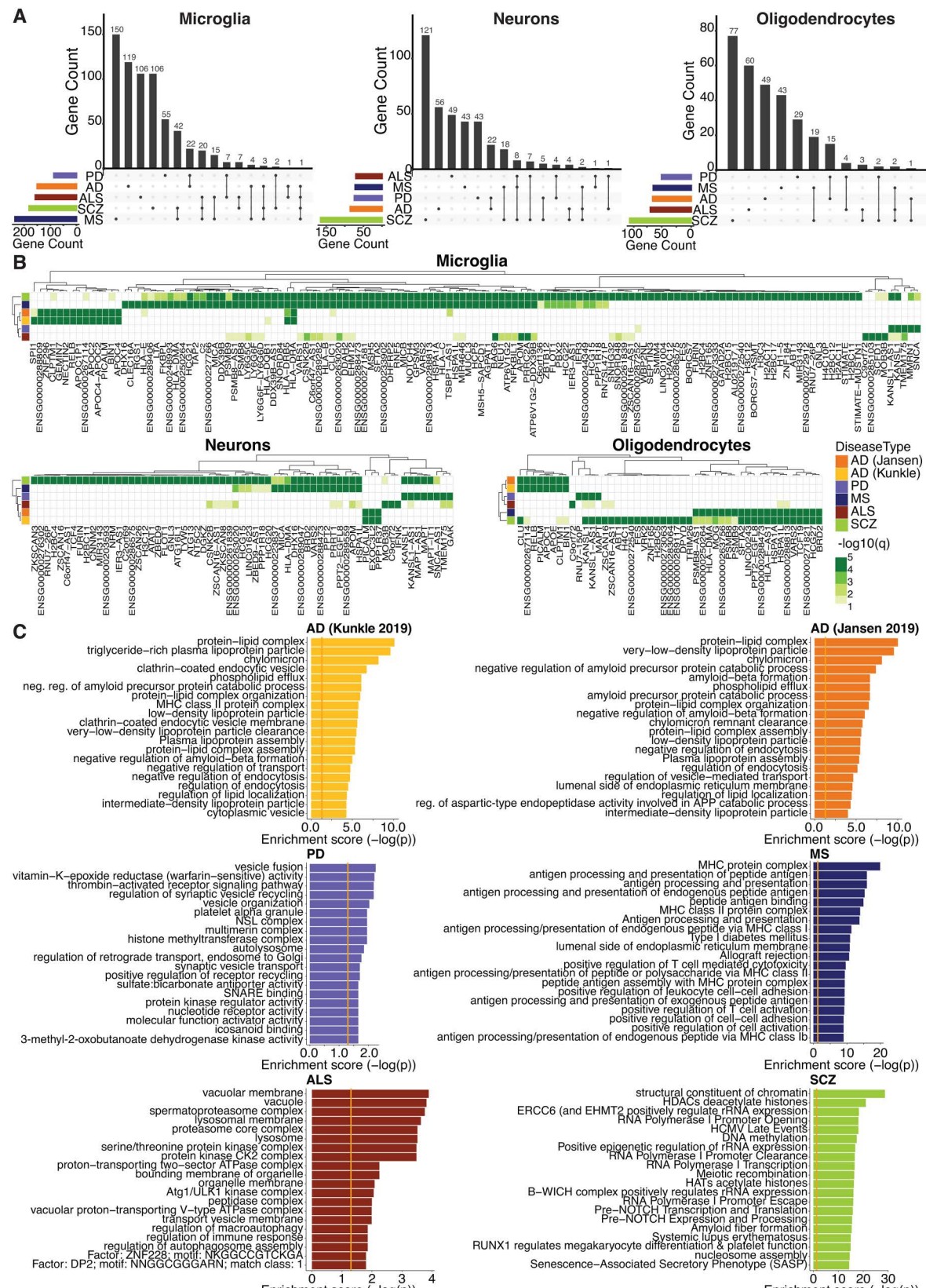

**Fig 3. Microglia disease-risk genes impacted disease-specific pathways.** A) UpSet visualization of unique and intersecting H-MAGMA disease-risk gene numbers between AD, PD (excluding 23andMe), MS, ALS and schizophrenia for each cell type. B) Heatmaps of H-MAGMA identified risk genes based on enhancer-to-promoter interactions from PLAC-seq data for AD, PD (excluding

23andMe), MS, ALS (H-MAGMA p<5e-8) and schizophrenia (H-MAGMA p<5e-12) for microglia, neurons and oligodendrocytes. Shown are H-MAGMA FDR corrected p-values (-log10(q)). C) Gene ontology pathway analysis of microglial risk genes identified by H-MAGMA for AD, PD (excluding 23andMe), MS, ALS, and schizophrenia; shown are top 20 pathways. SCZ, schizophrenia.

sharing pathways with PD related to lysosomes and autophagosomes (Fig 3c and S2 Table). Microglial-associated GO pathways for schizophrenia H-MAGMA-risk genes were distinct from the neurodegenerative disorders and primarily included epigenetic and gene regulatory pathways (Fig 3c and S2 Table). Neuronal GWAS schizophrenia risk genes were primarily implicated in synaptic processes (S5 Fig). The only common microglial-associated GO pathways between neurodegenerative disorders were autolysosome, cytoplasmic vesicle, secondary lysosome, vacuole and intracellular vesicle (S7 Fig). However, the combination of genes contributing to these pathways varied between diseases, with only HLA−DRB1/HLA−DRA locus shared for the autolysosomal and secondary lysosome pathways in ALS and AD (S7 Fig). Collectively, pathway analysis confirmed the observation from gene set overlaps, indicating that microglial risk genes and associated biological pathways are mostly disease specific.

## Discussion

Incorporation of enhancer-to-promoter interactomes for microglia, neurons and oligodendrocytes with GWAS summary statistics enabled us to identify the cell types and genes associated with the genetic risk of brain disorders. Partitioned heritability analysis highlighted microglia as an important cell type underlying genetic susceptibility across multiple neurodegenerative conditions. Accordingly, enhancer-to-promoter interactomes identified the greatest number of predicted risk genes in microglia for AD, PD, MS and ALS. Previous studies have shown both the importance of active regulatory regions [61,62] and that AD GWAS-risk is associated with gene regulatory regions in microglia [40–42,56], as well as monocytes and macrophages [43,63]. MS is an autoimmune condition where the immune system attacks the myelin sheath surrounding neurons [64], and MS genetic risk genes have been associated with the peripheral immune system and microglia [65]. ALS is a motor neuron disease that has been linked to aberrant inflammation [66], although GWAS risk for ALS has been primarily attributed to neuronal cell types [31]. The genetic risk of PD using single-cell gene expression analysis has identified dopaminergic neurons and oligodendrocytes as cell types that express PD risk genes [67,68]. Interestingly, PD GWAS risk was found to be enriched in microglia and monocyte chromatin accessibility regions [69], although equivalent epigenetic datasets for dopaminergic neurons are lacking. In summary, chromatin interactions in microglia showed the strongest heritability enrichment and revealed the most risk genes across all neurodegenerative disorders. Despite this commonality, microglia genetic-susceptibility genes identified using H-MAGMA were associated with pathways that were disease-specific. A previous study using LDSC to assess genome-wide genetic correlations (rg) found a low correlation between neurodegenerative GWAS datasets, with nominally significant positive associations between AD-ALS (0.19), AD-PD (0.23), and PD-ALS (0.23) [70]. This aligns with our findings that microglial regulatory regions enriched in disease are mostly disease specific.

Disease risk genes identified in microglia across neurodegenerative conditions exhibited a microglia-specific expression profile, whereas neuron- and oligodendrocyte-associated genes lacked consistent cell-type specificity (Fig 2d). This may reflect the distinct developmental origin of microglia from yolk sac progenitors, unlike neurons and oligodendrocytes, which arise from neural progenitor cells within the brain [71–73]. Furthermore, AD genetic risk in microglia was most strongly associated with phagocytic, inflammatory, and cycling microglial subtypes. Consistent with this, AD pathology has been linked to impaired phagocytosis,

heightened inflammatory activation, and a reduced microglial subpopulation enriched for cell cycle and DNA repair genes [74–76].

Pathway analysis of microglial AD H-MAGMA risk genes was associated with lipoproteins, amyloid processing, endocytosis and MHC class II. The lipid-protein complex and lipoprotein pathways included *APOE*, the strongest common genetic determinant of sporadic AD [77,78]. Microglia H-MAGMA genes previously linked to amyloid processing pathways included the ABC transporter *ABCA7*, vesicle-associated genes such as *PICALM*, *BIN1* and *SORL1*, and protein cleavage genes such as *ADAM10* and *APH1B* [79]. *PICALM* and *BIN1* have been further implicated in AD through microglial eQTL colocalization analyses [80] and *ABCA7* and *SORL1* through rare loss-of-function variants [81,82]. Further corroborating H-MAGMA in prioritizing AD risk genes, we identified genes associated with the *HLA* locus, immune genes (*INPP5D*) and microglia mobility (*ADAMTS4*, *CASS4*) that have previously been linked to AD genetic risk [28]. H-MAGMA also identified genes located more distal from AD GWAS variants that were not found using standard mapping strategies, including endosome/endocytosis-associated genes (*USP6NL*, *CNN2*, *RIN3*, *RAB8B*), membrane-associated genes (*SPPL2A*, *STX4*, *TRIP11*), gene regulation (*EED*) and protein homeostasis (*ATXN3*).

PD risk genes in microglia were associated with endo-lysosomal pathways, as previously implicated in a non-cell-type-centric manner for PD [83]. These included lysosomal-associated genes (*LRRK2*, *RAB29*, *PLEKHM1*), membrane fusion genes (*STX4*, *TMEM175*, *VPS37A*, *SNCA*) and endocytosis (*ARHGAP27*). The microglia PD risk genes *LRRK2*, *SNCA* (alpha-synuclein)*, and *TMEM175* have also been linked to rare coding mutations in PD patients [84–86]. The H-MAGMA-risk genes *NDUFAF2*, *BCKDK*, and *SUCNR1* have been associated with mitochondrial dysfunction, with *BCKDK* specifically linked to Parkinsonism and Maple Syrup Urine Disease (MSUD) [87,88], reinforcing evidence of mitochondrial impairment from early-onset PD mutations (*PINK1*, *PARK7*) and environmental factors [89,90]. Genes related to histone modifications were also identified as H-MAGMA-risk genes in microglia including *SETD1A* and *FAM47E* (lysine methylation), along with *KAT8* and *KANSL1* (lysine acetylation), which were also previously annotated as PD GWAS genes [8,27]. Inclusion of interactome data identified more distal PD H-MAGMA-risk genes in microglia, including *P2RY12* and *P2RY13* (immune homeostasis, mobility) [69] and *UBE2Q1* (stress response, cell death) [91]. H-MAGMA also highlighted *FGF20*, which supports dopamine secretion and protects nigral cells in PD models [92,93], highlighting the importance of distal microglial enhancers for these genes in PD.

MS H-MAGMA-risk genes were mostly associated with T cell signaling and antigen presentation and processing, consistent with previous findings [94]. A broader set of *HLA* genes were implicated in MS risk, as well genes not identified for AD including ABC transporter-related genes (*TAP1*, *TAP2*, *TAPBPL*) and MHC Class I and Class II-associated genes (*MICB*, *CIITA*). Additional risk genes implicated in antigen processing were heat shock proteins (*HSPA1B*, *HSPA1A* and *HSPA1L*) and the ubiquitin ligase *MARCHF1*. MS-risk genes associated with immune activation included Tumor Necrosis Factor (*TNF*) and TNF receptor family members *TNFRSF1* and *CD27,* negative regulation of cytokines (*SOCS1* and *VSIR*), interleukin signaling (*IL12RB1*) as well as other immune signaling molecules such as *AIF1* (also known as IBA1), *BCL10* and *PTPRC*. Interestingly, several chromatin-related risk genes were identified including *CORO1A* and the lysine acetyltransferase *KAT5*.

Pathways for ALS risk genes were mostly associated with vacuole-related terms, as well as autophagy and the lysosome. These included vacuole-associated channels and transporters *ATXN3* (spinocerebellar ataxia-3), *CLCN3*, *SLC12A4*, *TMEM175* and lysosomal-associated proteins such as *TPP1*, *KICS2*, *NEU1*, *TM6SF1*, as well as the guanine nucleotide exchange factor *C9orf72*, iduronidase *IDUA*, formin binding protein *FNBP1* and the vacuolar ATPase

*ATP6V1G2*. The proteasomal genes *PSMB8*, *PSMB9* and *PSMB10* were identified as MS-risk genes, with an isoform of *PSMB8* being linked to P-body formation in MS lesions [95]. Several kinases were identified besides *C9orf72*, including *TBK1* and *CSNK2B*. Repeat expansions in C9orf72 and mutations in *TBK1* have established associations with both ALS and frontotemporal dementia (FTD) [96,97].

Both sLDSC and H-MAGMA have aided in the systematic functional prediction of non-coding variants across complex traits and cell types, however, limitations of these approaches should be considered. Differences in imputation quality between GWAS summary statistics may introduce technical variability bias into the H-MAGMA and LDSC results. Due to the confounding effects of linkage disequilibrium in GWAS, some of the H-MAGMA identified risk genes may result in false positives. Additionally, the directionality of expression of the identified risk genes cannot be determined but can be partially addressed by incorporating cell type eQTL data [52,53,98]. Both H-MAGMA and sLDSC rely on the GWAS sample sizes and high SNP-heritability for accuracy of outcome [98,99]. In H-MAGMA the number of detected risk genes is dependent on the GWAS sample size and quality of chromatin interaction data. Therefore, not all identified risk genes are truly associated with the trait, necessitating functional validation. Both sLDSC and H-MAGMA were applied to European-ancestry populations due to limiting availability of large sample-sized GWAS, LD reference panel and custom genotyping arrays for other ancestries. Epigenetic and chromatin interaction data derived from pediatric resected tissue peripheral to epileptic foci was used to mitigate against aging-associated reverse causality. However, epilepsy may influence cell types more broadly throughout the brain. Incorporating epigenomic datasets spanning different ages and conditions as these become available will offer deeper insights into GWAS loci that may be context specific. The sLDSC and H-MAGMA analysis was focused on enhancer-to-promoter interactions, given their functional significance in neurodegenerative disorders [46,61,62]. However, PLAC-seq interactions anchored outside of promoters and enhancers ('other interactions') potentially represent insulator- and repressor-associated chromatin loops [92,100] that may have additional functional implications on genetic risk.

The assignment of cell types to genetic risk and the identification of target genes depends on cell type epigenomic and chromatin interactome profiling. This has been performed for a limited number of cell types and chromatin conformation data has mostly been generated for non-dementia cases, which limited the analysis to microglia, neurons and oligodendrocytes. Recent gene expression studies have implicated vascular cell types in the genetic risk for AD [58,101,102]. Furthermore, the expression of AD risk has been reported to be differentially enriched in microglia substates [59]. These examples highlight the need for epigenomic and chromatin conformation analysis of rare cell types and substates across disease conditions using single-cell approaches. However, our current analysis reinforces the genetic causative role of microglia in age-related brain conditions and offers biological insights into their involvement in various neurodegenerative disorders.

## Materials and methods

### PLAC-seq datasets

PLAC-seq data for human microglia, neurons and oligodendrocytes was pre-processed by Nott et al., 2019 [40]. PLAC-seq data was generated using epilepsy resections of the frontal, parietal and temporal cortex of seven individuals aged 5 months to 17 years of Caucasian, Hispanic and Pacific Islander ethnicities (6 males/1 female). PLAC-seq chromatin interactions were anchored to active gene promoters by immunoprecipitation of histone H3 lysine 4 trimethylation (H3K4me3), which is a histone modification enriched at active gene promoters

[54,55]. Chromatin interactions were 5 kb resolution and were anchored to promoters using chromatin immunoprecipitation of the histone modification H3K4me3 [40].

## Classification of PLAC-seq interactions

PLAC-seq chromatin interactions were classified as i) promoter-to-enhancer; ii) promoter-to-promoter; iii) promoter-to-ATAC; iv) promoter-to-promoter/enhancer; v) promoter-to-other; vi) H3K4me3-to-H3K4me3; vii) H3K4me3-to-other; and viii) other interactions for microglia, neurons and oligodendrocytes. 'Promoter' were classified as PLAC-seq bins that overlapped with H3K4me3 and H3K27ac regions within 2,000 bp of the nearest TSS and 'enhancer' were classified as PLAC-seq bins that overlapped H3K27ac regions distal to TSS as defined by Nott 2019 [40]; promoter/enhancer were classified as PLAC-seq bins that overlapped both promoter and enhancer regions; 'H3K4me3' were PLAC-seq bins that overlapped H3K4me3 regions distal from TSS; 'ATAC' were PLAC-seq bins that overlapped chromatin accessibility regions that were devoid of H3K4me3 and H3K27ac; 'other' were PLAC-seq bins that did not overlap with H3K4me3, H3K27ac or chromatin accessibility regions [40]. To identify the number of enhancers interacting with each promoter and number of promoters interacting with each enhancer, cell type PLAC-seq bins were overlapped with active promoter and active enhancer regions.

## GWAS datasets

The following GWAS summary statistics were used in this study were downloaded from EBI's GWAS catalogue (https://www.ebi.ac.uk/gwas/) and were of European ancestry:

AD (Jansen 2019) (GCST007320): n= 71,880 cases and 383,378 controls [29];

AD (Kunkle 2019) (GCST007511): n = 21,982 cases and 41,944 controls, Stage 1 [30];

PD (Nalls 2019) (GCST009325): n = 33,674 cases and 449,056 controls (excluding 23andMe) [27];

MS (Andlauer 2016) (GCST003566): n = 4,888 cases and 10,395 controls [33];

ALS (van Rheenen 2021) (GCST90027164): n = 27,205 cases and 110,881 controls [31];

schizophrenia (Trubetskoy 2022) (GCST90128471): n = 53,386 cases and 77,258 controls [32].

## Quality control of GWAS summary statistics

GWAS summary statistics were standardised and underwent quality control steps before running H-MAGMA. GWAS summary statistics were filtered using format_sumstats function in "MungeSumstats" package (version 1.6.0, available on Bioconductor) in R (version 4.2.1) [103]. Quality control steps included SE > 0, SNPs that have N<5 standard deviation above mean, indels were kept, chromosomes X, Y and mitochondrial chromosome were removed and multiallelic SNPs were kept. GWAS summary statistics had the following imputation quality scores: AD (Jansen 2019) >0.91; AD (Kunkle 2019) >0.4; PD >0.8; MS ≥0.8; ALS >0.95; schizophrenia >0.9. The pre-filtered imputation quality scores provided by the original GWAS were used for AD, PD, MS and ALS. For schizophrenia, we used an imputation quality score that matched the AD GWAS (Jansen 2019).

## H-MAGMA

Annotating genetic variants to target genes was performed using H-MAGMA workflow [52,98]. H-MAGMA workflow incorporates 1) generating input files which provide the background profile of gene-SNP associations based on chromatin interaction data and 2) using MAGMA software to run gene-level analysis (S2 Fig). To generate cell type-specific

promoter-enhancer profiles, 1) chromatin interaction data from PLAC-seq for microglia, neurons and oligodendrocytes, and 2) reference data for SNPs (22665064 SNPs) from Phase 3 of 1,000 Genomes for European ancestry were used (genome Build 37) (https://cncr.nl/research/magma/). Chromatin interaction data from PLAC-seq with FDR-corrected p-value cut-off of 0.01 were filtered to interactions with promoters in at least one end by overlapping with cell-type promoter regions to only investigate active chromatin interactions [40]. Gene-SNP association profiles were created by assigning exonic and promoter SNPs directly to target genes based on genomic location using a gene model Gencode v41 [104]. Promoters were defined as 1.5kb upstream and 500 bp downstream of the TSS of each gene isoform based on prior publications [105,106]. Intronic and intergenic SNPs were assigned to genes based on cell-type chromatin interactions (see PLAC-seq datasets) with promoters and exons [52]. Intronic and intergenic SNPs were filtered to enhancer SNPs by overlapping with cell-type enhancer regions [40]. To investigate disease enrichment in active chromatin interactions, significant cell-type specific chromatin interactions with FDR-corrected p-value cut-off of 0.01 were filtered to interactions with promoters in at least one end by overlapping cell-type promoter regions [40]. Filtered chromatin interactions were overlapped with Gencode 41 exon and promoter coordinates to identify exon-based and promoter-based interactions [52,98].

To determine whether enhancer or promoter/exon SNPs were driving the disease enrichment of genes, H-MAGMA input files were generated either with promoter/exon SNPs or enhancer SNPs only. Exonic and promoter SNPs were directly assigned to target genes based on genomic location using a gene model Gencode v41 (https://www.gencodegenes.org/human/release_41lift37.html) [104]. Promoters were defined as 1.5kb upstream and 500 bp downstream of the TSS of each gene isoform. Intronic and intergenic SNPs were assigned to 'ALL' Gencode v41 genes based on cell-type chromatin interactions (see PLAC-seq datasets) with promoters and exons [52]. Intronic and intergenic SNPs were filtered to enhancer SNPs by overlapping with cell-type enhancer regions [40]. After running gene-level analysis using MAGMA software (see MAGMA section below) outputted genes with an FDR-corrected p-value <0.05 were selected for downstream analysis (https://ctg.cncr.nl/software/magma).

## MAGMA

MAGMA analysis pipeline was used to run the cell type-specific gene level association with a disease based on gene-SNP profiles based on chromatin interaction data [53]. The association was established using the default "SNP-wise mean" gene analysis model, which is a test of mean SNP association using the sum of squared SNP Z-statistics as a test statistic. In brief, SNP-level p-values from GWAS summary statistics were aggregated into gene-level p-values and a reference data set (1,000 Genomes European panel) was used to account for linkage disequilibrium between SNPs. Since some of the GWAS summary statistics used in the study are SNP meta-analysis results, individual sample sizes per SNP may have significant variation and may affect the gene test-statistic results. Therefore, if available, individual sample sizes per SNP were used (ncol modifier in –pval parameter in MAGMA). The analysis was run as follows: magma --bfile g1000_eur --pval <GWAS summary statistics> use=SNP,P ncol=NSUM --gene-annot <Input annotation file> --debug set-spar=tmp_snps_used --out <Output file>

Standard gene mapping with MAGMA was applied to assign risk genes for each disease without incorporating epigenetic or chromatin interaction datasets, with variants assigned to genes based on proximity within a 10 kb window. The analysis was run as follows: 1) magma --annotate window=10,10 --snp-loc g1000_eur.bed --gene-loc gencode41_gene_coord.txt --out <Output Annotation>; 2) magma --bfile g1000_eur --pval <GWAS summary statistics> use=SNP,P ncol=NSUM --gene-annot <Input annotation file> --debug set-spar=tmp_snps_used --out <Output file>

## Partitioned heritability (sLDSC regression)

Partitioned heritability using sLDSC regression analysis was used to identify brain cell type annotations that were enriched for heritability of AD, PD (excluding 23andMe), MS, ALS and schizophrenia (LDSC version 1.0.1) by functional category while controlling for 97 annotation categories of the full baseline model (model version 2.2) [99]. Cell type annotations per functional category were run jointly. Functional categories included cell type 1) total PLAC-seq bins, 2) promoter and enhancer PLAC-seq bins, 3) promoters PLAC-seq bins, and 4) enhancer PLAC-seq for microglia, neurons and oligodendrocytes. Baseline model LD scores, standard regression weights, and allele frequencies that were used were built from 1000 Genomes Phase 3 for European population. The enrichment P-values were FDR multiple testing corrected for the number of GWAS studies and number of cell types using Benjamini-Hochberg correction method. Disease enrichment was considered insignificant if the coefficient z-score was negative. Cell type annotations for all the functional categories were created using plink format.bed/.bim/. fam filesets of 1000 Genomes Phase 3 for European population and LD scores were computed based on a 1 centiMorgan (cM) window. Since the annotations were built on top of the baseline model, 1000 Genomes Phase 3 was used together with the HapMap3 SNPs. A quality control step of GWAS summary statistics was performed before LDSC analysis using munge_sumstats. py where SNPs had INFO <= 0.9, MAF <= 0.01 and N < 32290, were out-of-bounds p-values, strand-ambiguous, with duplicated IDs and alleles did not match Hap-Map SNPs. To prevent bias from variable imputation quality both between and within each GWAS study, all the GWAS SNPs were filtered to HapMap3 SNPs, as these SNPs are well imputed in most studies.

## EWCE

Expression weighted cell type enrichment (EWCE) analysis (v1.6.0) was used to identify cell type-specificity of the H-MAGMA outputted risk genes for each disease type [107]. Single-cell RNA-seq data from brain cortical tissue from Tsartsalis et al. (2014) [58] was used to generate probability distribution associated with cell type-specific H-MAGMA outputted risk genes having an average level of expression within a cell type of the brain. Single-cell RNA-seq data for human microglial subtypes from Sun et al. (2023) [59] was used to generate probability distribution associated with microglia H-MAGMA outputted risk genes having an average level of expression within the subtypes. Significant cell type-specificity was determined based on the p-value <0.05.

## GO analysis

Gene set enrichment analysis was performed on the list of H-MAGMA outputted significant risk genes identified per cell type to identify biological pathways at risk in each cell type for each disease. The R package "gprofiler2" (v0.2.1) was used for gene set enrichment, which contains data sources including Gene Ontology (GO), KEGG, Reactome, WikiPathways, miRTarBase, TRANSFAC, Human Protein Atlas, protein complexes from CORUM and Human Phenotype Ontology [108]. Risk genes inputted into the analysis were filtered based on the FDR adjusted p-value<0.05 and were ordered based on the Z-score generated by the H-MAGMA. Identified pathways were also FDR corrected using p-value <0.05. For visualization, if pathways contained the same set of genes, the one with the highest FDR corrected p-value was included in the bar plots.

## Supporting information

**S1 Fig. LDSC coefficient z-scores and enrichment values.** A) Partitioned heritability sLDSC coefficient z-scores for i) total PLAC-seq bins (ii) promoter and enhancer PLAC-seq bins; iii) all promoters and iv) all enhancers for microglia, neurons and oligodendrocytes in AD, PD (excluding

23andMe), MS, ALS, and schizophrenia. *transformed coefficient p-values < 0.05. **B)** Partitioned heritability sLDSC enrichment values defined as the ratio of the proportion of heritability to the number of SNPs (Prop. h2/ Prop. SNPs) for i) total PLAC-seq bins (ii) promoter and enhancer PLAC-seq bins; iii) all promoters and iv) all enhancers for microglia, neurons and oligodendrocytes in AD, PD (excluding 23andMe), MS, ALS, and schizophrenia. The grey dotted line represents the cutoff for enrichment (1). Error bars represent standard error. SCZ, schizophrenia.
(TIF)

**S2 Fig.  H-MAGMA workflow.** Step 1, genes are annotated to their associated SNPs using chromatin interaction data, enhancers, and promoters that were derived from each cell type. Step 2, gene analysis is performed using MAGMA software and relevant disease phenotypes (GWAS summary statistics) to identify disease-associated risk genes.
(TIF)

**S3 Fig.  Comparison of risk genes identified by MAGMA and H-MAGMA.** Venn diagrams (left) illustrate the overlap between genes identified by MAGMA (green) and microglia-specific H-MAGMA (blue) across diseases for: A) AD (Jansen et al. 2019) [29], B) AD (Kunkle et al., 2019)[30], C) PD (minus 23andme), D) MS, and E) ALS. Dot plots (middle) display the top 20 genes identified exclusively by H-MAGMA. Bar plots (right) represent the top pathways containing at least one of these H-MAGMA-unique genes.
(TIF)

**S4 Fig.  H-MAGMA disease risk genes identified using PLAC-seq interactions overlapping SNPs subset to either genes or enhancers only.** A) The number of disease-risk genes identified in microglia, neurons and oligodendrocytes using H-MAGMA and GWAS for AD, PD (excluding 23andMe), MS, ALS, and schizophrenia using SNPs overlapping PLAC-seq bins at i) exon and promoters only (left) or at ii) enhancer regions only (right). B) Chromatin interactions were randomly sampled down 10 times to 60,000 interactions and the number of disease-risk genes were identified in microglia, neurons and oligodendrocytes using H-MAGMA and GWAS for AD, PD (excluding 23andMe), MS, ALS, and schizophrenia using SNPs overlapping PLAC-seq bins at i) exons and promoters only (left) or at ii) enhancers only (right). Dunn's test (non-parametric) between cell types within each group: i) exons and promoters only: AD (Jansen 2019): microglia-oligodendrocytes (**), PD: microglia-oligodendrocytes (**), neurons-oligodendrocytes (***); MS: microglia-neurons (****), microglia-oligodendrocytes (**); ALS: microglia-neurons (*), microglia-oligodendrocytes (****), neurons-oligodendrocytes (*); schizophrenia: microglia-neurons (****), microglia-oligodendrocytes (*****) and ii) enhancers only: AD (Jansen 2019): microglia-neurons (****), microglia-oligodendrocytes (**); PD: microglia-neurons (**), microglia-oligdendrocyteso (****), neurons-oligodendrocytes (*); MS: microglia-neurons (****), microglia-oligodendrocytes (**); ALS: microglia-neurons (**), neurons-oligodendrocytes (****); schizophrenia: microglia-neurons (*), microglia-oligodendrocytes (*), neurons-oligodendrocytes (****). *p <0.05, **p<0.01, ***p<1e-4, ****p<1e-6. SCZ, schizophrenia.
(TIF)

**S5 Fig.  Gene ontology pathways for neurons across diseases.** Gene ontology pathway analysis of neuronal risk genes identified by H-MAGMA for AD, PD (excluding 23andMe), MS, ALS, and schizophrenia; shown are the top 20 pathways. SCZ, schizophrenia.
(TIF)

**S6 Fig.  Gene ontology pathways for oligodendrocytes across diseases.** Gene ontology pathway analysis of oligodendrocyte risk genes identified by H-MAGMA for AD, PD

(excluding 23andMe), MS, ALS, and schizophrenia; shown are the top 20 pathways. SCZ, schizophrenia.
(TIF)

**S7 Fig. Overlap of microglial pathways across diseases and their associated genes.** A) Number of overlapping microglial pathways across diseases. B) Bar plots of the five microglial pathways shared between AD, PD, MS, and ALS. The y-axis represents genes associated with each pathway per disease, the x-axis indicates pathway significance p-value (–log10(p-value), in relation to the disease, and colours denote different diseases.
(TIF)

**S1 Table. H-MAGMA risk genes.** Risk genes identified by H-MAGMA using microglia, neuron and oligodendrocyte chromatin interaction data and GWAS for neurodegenerative conditions and schizophrenia.
(XLSX)

**S2 Table. Gene ontology pathways for H-MAGMA genes.** Gene ontology analysis of risk genes identified by H-MAGMA in microglia, neurons and oligodendrocytes for neurodegenerative conditions and schizophrenia.
(XLSX)

**S3 Table. MAGMA risk genes.** Risk genes identified by MAGMA using 10kb regions flanking all the genes and GWAS for neurodegenerative conditions and schizophrenia.
(XLSX)

**S4 Table. H-MAGMA exclusive risk genes.** Risk genes identified by H-MAGMA but not MAGMA across microglia, neurons and oligodendrocytes for neurodegenerative conditions and schizophrenia.
(XLSX)

**S5 Table. Gene ontology pathways for H-MAGMA exclusive genes.** Gene ontology analysis of risk genes identified by H-MAGMA only (not MAGMA) in microglia, neurons and oligodendrocytes for neurodegenerative conditions and schizophrenia.
(XLSX)

**S6 Table. Shared pathways between cell types per disease based on H-MAGMA genes.**
(XLSX)

## Acknowledgements

We thank Dr Hyejun Won for the discussions on H-MAGMA. We thank the Skene, Marzi, Johnson and Nott groups for advice and helpful discussion, in particular, Dr Alan Murphy, Dr Brian Schilder, Dr Kitty Murphy and Dr Alex Haglund.

## Author contributions

**Conceptualization:** Alexi Nott.

**Data curation:** Aydan Askarova, Reuben M Yaa.

**Formal analysis:** Aydan Askarova.

**Investigation:** Aydan Askarova.

**Methodology:** Aydan Askarova, Alexi Nott.

**Project administration:** Aydan Askarova.

**Resources:** Alexi Nott.

**Software:** Sarah J Marzi.

**Supervision:** Reuben M Yaa, Sarah J Marzi, Alexi Nott.

**Visualization:** Aydan Askarova.

**Writing – original draft:** Aydan Askarova, Alexi Nott.

**Writing – review & editing:** Reuben M Yaa, Sarah J Marzi.

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
