## [Decision Letter · Decision Letter 0]

24 Dec 2024

PGENETICS-D-24-00990

Genetic risk for neurodegenerative conditions is linked to disease-specific microglial pathways

PLOS Genetics

Dear Dr. Nott,

Thank you for submitting your manuscript to PLOS Genetics. After careful consideration, we feel that it has significant merit but does not fully meet PLOS Genetics's publication criteria as it currently stands. Therefore, we invite you to submit a revised version of the manuscript that addresses the points raised during the review process.

Please submit your revised manuscript within 60 days Feb 22 2025 11:59PM. If you will need more time than this to complete your revisions, please reply to this message or contact the journal office at plosgenetics@plos.org. Please include the following items when submitting your revised manuscript:

We look forward to receiving your revised manuscript.

Kind regards,

Annie Vogel Ciernia

Guest Editor

PLOS Genetics

Wendy Bickmore

Section Editor

PLOS Genetics

Aimée Dudley

Editor-in-Chief

PLOS Genetics

Anne Goriely

Editor-in-Chief

PLOS Genetics

**Journal Requirements:**

At this stage, the following Authors/Authors require contributions: Aydan Askarova, Reuben M Yaa, Sarah J Marzi, and Alexi Nott. Please ensure that the full contributions of each author are acknowledged in the "Add/Edit/Remove Authors" section of our submission form.

The list of CRediT author contributions may be found here: https://journals.plos.org/plosgenetics/s/authorship#loc-author-contributions

https://journals.plos.org/plosgenetics/s/submission-guidelines#loc-parts-of-a-submission

5) We notice that your supplementary Figures are included in the manuscript file. Please remove them and upload them with the file type 'Supporting Information'. Please ensure that each Supporting Information file has a legend listed in the manuscript after the references list.

6) Please ensure that the funders and grant numbers match between the Financial Disclosure field and the Funding Information tab in your submission form. Note that the funders must be provided in the same order in both places as well. State the initials, alongside each funding source, of each author to receive each grant. For example: "This work was supported by the National Institutes of Health (####### to AM; ###### to CJ) and the National Science Foundation (###### to AM)." State what role the funders took in the study. If the funders had no role in your study, please state: "The funders had no role in study design, data collection and analysis, decision to publish, or preparation of the manuscript.".

**Reviewers' comments:**

Reviewer's Responses to Questions

**Comments to the Authors:**

Reviewer #1: This is an interesting manuscript, which addresses gaps in our knowledge relating to the cell-specific functions of non-coding variants identified from GWAS of neurodegenerative disorders (Alzheimer’s disease, Parkinson’s disease, multiple sclerosis, and amyotrophic lateral sclerosis). To address these knowledge gaps, the authors harnessed information from enhancer-to-promoter interaction maps for microglia, neurons and oligodendrocytes to identify cell type specific enrichments for these different neurodegenerative disorders. From these analyses, the authors show that while microglia were identified as the most important cell types for neurodegenerative disorders, the genes and pathways that are affected in these cells are largely unique for each of the examined disorders. While this is an interesting study, this manuscript would be strengthened by addressing the comments listed below:

The abstract would benefit from including mention to the results from the analyses of the schizophrenia GWAS.

P4, lines 105-115. There are a few areas in the results where there is repetition of the methods. This should be checked throughout.

P5, line 122-124. Please provide the range in addition to the average distance for each cell type.

P6, line 162. The “Microglial chromatin interactions identify disease risk genes across multiple neurodegenerative conditions” section should include a comparison of how many disease genes would be identified by using standard gene mapping strategies that don’t don’t consider cell-specific interactions with distal gene regulatory regions. How many more genes are identified using the PLAC-seq data that would not have been identified as disease genes using standard methods? What are these genes?

P7, lines 192-201. It would be helpful to provide numbers for the disease risk genes in the main text so readers don’t need to refer to Supplementary data.

P8, line 216-220. This text would fit better in the discussion.

Figure 2. Use the same scales for the graphs of neurodegenerative disorders to allow for easier comparison between the diseases. Check this throughout. Also some comment should be made on why there are so many more disease genes identified for schizophrenia and how this may impact other results reported on in the paper.

Figure 3B is very difficult to read. Gene names should be provided in a consistent format.

P10, line 271-273. Comparison to results obtained not making use of these interactome data would be helpful in the discussion.

The discussion of the disease genes that were identified should be shortened and focus on genes that have not been identified through previous GWAS fine-mapping/annotation analyses i.e., those that were uncovered through the inclusion of PLAC-seq data as this is the major focus of this study. What novel information has this study provided in terms of the identification of disease genes.

The discussion would benefit from some comment on why certain analyses did not uncover any significant results for certain disorders. The addition of LDSC heritability and correlation analyses would be helpful to put into context how closely the investigated disorders are related to each other.

Some discussion should be provided on why “other” interactions are so frequently observed.

Some comment on the limitations should also be made. For example, the authors only looked at three cell types. Other cell types could also be of importance, therefore future studies would benefit from the examination of other cell types.

P26, line 501. Information is provided on age ranges, but it would also be helpful to comment on the sex and ancestry/ethnicity of these samples as all GWAS cohorts were of European ancestry. These data were also obtained from epilepsy resections. Some comment on how the use of these samples may influence results looking at other disorders.

P26, line 521. Information on why these specific GWAS phenotypes were selected should be included? The authors should comment on how GWAS sample sizes may impact power to uncover variants associated with the respective diseases and how this in turn may impact downstream analyses.

P27, line 537-538. Why were different imputation quality scores applied to the different summary stats? Some information on how variant overlap between the summary statistics was assessed would also be helpful (i.e., was the number of variants included, reference populations for imputation, QC measures similar)?

P27, line 545. Number of SNPs included from Phase 3 1000GP should be corrected.

P27, line 550. Why is the definition of promoter changed from 2,000 bp from TSS on p26, to 1,500 bp from TSS here?

P27, line 551. Provide information on how cognate genes were defined.

P27, line 540. The H-MAGMA section is difficult to follow and would benefit from the inclusion of a figure visualizing how genetic variants were mapped to target genes and how disease enrichment analyses were performed.

P29, line 604. Why were these specific mouse single cell data selected?

Reviewer #2: Here the authors examined common variants (GWAS) in multiple neurodegenerative diseases (NDs). It has always been a challenge to ascribe meaning to GWAS hits as most are not in coding regions. They hypothesize that since many GWAS hits are enriched at chromatin accessibility regions that are likely enhancers or promoters that histone modifications may play a large role where cell-type specificity is at play. Hence they form the research question of what are the target genes for these GWAS hits? And what cell-types. They use new computational tools, namely, H-MAGMA (Hi-C coupled multimarker analysis of genomic annotation) and found an enrichment of unique microglial genes in AD, PD, ALS, and MS. They used PLAC-seq to show microglia enhancer to promoter interactions were enrichment for AD, PD, and MS. Fitting with previous reports, schizophrenia was enriched in neurons and oligos. They identified some shared microglial genes across NDs, but mostly there were specific microglial genes for the separate NDs. Moreover, specific gene ontology was found across the NDs.

The paper is filled with a wealth of analysis and is very well written.

My only suggestion is to help the functional modellers out. A figure 4 should be generated to highlight the top hits (10 genes or so) in what cell types that should be functionally addressed in future studies of the NDs and schizophrenia.

Reviewer #3: Reviewer Report for Manuscript: PGENETICS-D-24-00990

This manuscript investigates the genetic risk for neurodegenerative conditions by leveraging enhancer-to-promoter interactomes and GWAS data to highlight microglial pathways as critical components in disease etiology. The authors employ H-MAGMA, sLDSC, and EWCE analyses to identify cell-type-specific risk genes and associated pathways. Their findings provide compelling evidence for the role of microglia in neurodegenerative diseases, with specific attention to disease-relevant pathways.

The study aims to address an important topic in neurogenomics and offers valuable insights into the mechanisms underlying genetic susceptibility in neurodegenerative disorders mapping how immune processes display disease-specific patterns and identified putative cell types and genes that contribute to the genetic susceptibility of these disorders. Despite some limitations, the integration of multiple datasets and computational approaches is the greatest asset of the study. However, there are a couple of key areas where the study could benefit from some additional analyses and/or clarification to enhance its robustness and impact.

Major Comments

1.- The study would benefit from showing specific examples of the genomic regions of interest where GWAS loci overlap with PLAC-seq data. Including traces of contact frequencies or interaction loops within these regions would allow readers to better evaluate the quality and relevance of the raw data. Provide representative genome browser snapshots or visualizations (e.g., IGV or UCSC Genome Browser) highlighting GWAS regions, PLAC-seq interaction sites, and associated promoter regions. Annotating key features such as enhancer-promoter interactions, SNP locations, and nearby risk genes would make these relationships more tangible and strengthen the reader's confidence in the findings. To this reviewer, this addition would make the results more transparent and provide an intuitive way for readers to assess the data integration.

2.- In regards to H-MAGMA predictions: the reliance on H-MAGMA for assigning GWAS variants to genes is a strong point of this study. However, the manuscript would benefit from additional evidence supporting the predicted gene-disease associations. Specifically, linking identified microglial genes (e.g., APOE, PICALM, and LRRK2) to the pathways implicated in the GO analysis through previously published literature or further exploration of their dataset would strengthen the claims. I would suggest authors incorporating supporting evidence from the literature or data analyses to substantiate these associations. For example, highlight prior studies that connect these genes to the identified pathways or discuss how the current dataset aligns with or diverges from known findings.

3.- Differentiating Disease-Specific and Shared Pathways. While the analysis highlights disease-specific pathways for microglia, neurons, and oligodendrocytes, it is unclear how much overlap exists among pathways across cell types and conditions. For instance, some pathways appear in multiple contexts but may vary in significance. The study would benefit from including a figure or table (rather than describing specific genes or pathways in the text) explicitly quantifying shared and unique pathways across diseases and cell types to improve clarity.

4.- Microglial Subtypes and Disease Associations: Recent studies highlight the heterogeneity of microglial subtypes in disease contexts. The authors should comment on (or even explore) how specific microglial subtypes might be linked to their findings. As a suggestion, include references to single-cell RNA-seq datasets or discuss how integrating these datasets could refine subtype-specific risk gene associations.

Minor Comments

1) The study would benefit from authors touching on general limitations by:

a) Briefly outlining any biases or assumptions inherent in using H-MAGMA or sLDSC that may impact the findings.

b) Addressing sLDSC-Specific Limitations - noting how linkage disequilibrium patterns in GWAS studies might affect the resolution of enriched pathways or cell types, potentially leading to false positives or missed signals.

c) Acknowledging factors of computational nature: both H-MAGMA and sLDSC rely on computational predictions, which inherently lack experimental validation and depend on the quality of input datasets (e.g., GWAS, Hi-C).

d) Highlighting any challenges in integrating results from multiple computational approaches (e.g., discrepancies between H-MAGMA and sLDSC outcomes) and how these were addressed or interpreted.

2) Data Availability and Reproducibility: The authors provide links to code and processed data? but these links did not work for this reviewer. This issue is likely due to placeholder uploads or disruptions during the editorial process. However, as of this review, it remains unclear whether raw data or intermediate results (e.g., interaction matrices) are or will be readily accessible. Clear data availability is critical for ensuring reproducibility. Please verify that all datasets and scripts are accessible and include a more detailed description of the analysis pipeline to support transparency and reproducibility.

Conclusion

This manuscript aids in understanding the genetic underpinnings of neurodegenerative disorders by integrating genomic tools and datasets. While the findings are robust and timely, additional analyses and clarifications as highlighted above would significantly enhance the impact of the study. With these revisions, this work could make a valuable contribution to the field.

**Have all data underlying the figures and results presented in the manuscript been provided?**

Reviewer #1: Yes

Reviewer #2: Yes

Reviewer #3: **No: ** Links provided did not work for this reviewer. This was communicated to the authors within my review.

PLOS authors have the option to publish the peer review history of their article (what does this mean? ). If published, this will include your full peer review and any attached files.

**Do you want your identity to be public for this peer review?** For information about this choice, including consent withdrawal, please see our Privacy Policy .

Reviewer #1: No

Reviewer #2: No

Reviewer #3: No

**Figure resubmission:**
---

## [Decision Letter · Decision Letter 1]

19 Mar 2025

PGENETICS-D-24-00990R1

Genetic risk for neurodegenerative conditions is linked to disease-specific microglial pathways

PLOS Genetics

Dear Dr. Nott,

Thank you for re-submitting your manuscript to PLOS Genetics. Please address the remaining minor suggestions from reviewer 1 prior to publication. We invite you to submit a revised version of the manuscript that addresses these final points raised during the review process.

Please submit your revised manuscript within 30 days Apr 18 2025 11:59PM. If you will need more time than this to complete your revisions, please reply to this message or contact the journal office at plosgenetics@plos.org. Please include the following items when submitting your revised manuscript:

We look forward to receiving your revised manuscript.

Kind regards,

Annie Vogel Ciernia

Guest Editor

PLOS Genetics

Wendy Bickmore

Section Editor

PLOS Genetics

Aimée Dudley

Editor-in-Chief

PLOS Genetics

Anne Goriely

Editor-in-Chief

PLOS Genetics

**Additional Editor Comments :**

Please address remaining comments by reviewer 1 before publication.

**Reviewers' comments:**

Reviewer's Responses to Questions

Reviewer #1: The authors have carefully addressed the reviewers comments, which has strengthened the manuscript. There are only a few minor additions that are suggested as follows:

16. P27, line 537-538. Why were different imputation quality scores applied to the different summary stats? Some information on how variant overlap between the summary statistics was assessed would also be helpful (i.e., was the number of variants included, reference populations for imputation, QC measures similar)?

We have provided further information on the quality measures that were used in the ‘Quality control of GWAS summary statistics’ section (lines 579-585): “Quality control steps included SE > 0, SNPs that have N<5 standard deviation above mean, indels were kept, chromosomes X, Y and mitochondrial chromosome were removed and multiallelic SNPs were kept. GWAS summary statistics had the following imputation quality scores: AD (Jansen 2019) >0.91; AD (Kunkle 2019) >0.4; PD >0.8; MS ≥0.8; ALS >0.95; schizophrenia >0.9. The pre-filtered imputation quality scores provided by the original GWAS were used for AD, PD, MS and ALS. For schizophrenia, we used an imputation quality score that matched the AD (Jansen 2019) GWAS.”

Response: Thank you for providing these details. As the differences in imputation quality scores could impact the results obtained, some mention of these concerns should be made in the limitations section.

18. P27, line 550. Why is the definition of promoter changed from 2,000 bp from TSS on p26, to 1,500 bp from TSS here?

PLAC-seq bins that overlap a H3K4me3 and H3K27ac peak within 2,000 bp of a transcriptional start site (TSS) used promoter annotations previously defined by Nott et al 2019. Given that the PLAC-seq loops have a resolution of 5kb, a broader definition of promoters was appropriate. To subsequently assign SNPs to promoters and exons for the annotation file used by H-MAGMA to identify risk genes it was decided that a stricter criterion would be used based on previous publications (Molineris Mol Biol Evol. 2011; Nair Cell Genom. 2024).

Molineris I, Grassi E, Ala U, Di Cunto F, Provero P. Evolution of promoter affinity for transcription factors in the human lineage. Mol Biol Evol. 2011 Aug;28(8):2173-83. doi: 10.1093/molbev/msr027. Epub 2011 Feb 18. PMID: 21335606.

Nair VD, Pincas H, Smith GR, Zaslavsky E, Ge Y, Amper MAS, Vasoya M, Chikina M, Sun Y, Raja AN, Mao W, Gay NR, Esser KA, Smith KS, Zhao B, Wiel L, Singh A, Lindholm ME, Amar D, Montgomery S, Snyder MP, Walsh MJ, Sealfon SC; MoTrPAC Study Group. Molecular adaptations in response to exercise training are associated with tissue-specific transcriptomic and epigenomic signatures. Cell Genom. 2024 Jun 12;4(6):100421. doi: 10.1016/j.xgen.2023.100421. Epub 2024 May 1. PMID: 38697122; PMCID: PMC11228891.

Response: Thank you for providing these details. Please add these citations to the manuscript text for clarity.

20. P27, line 540. The H-MAGMA section is difficult to follow and would benefit from the inclusion of a figure visualizing how genetic variants were mapped to target genes and how disease enrichment analyses were performed.

We have provided a schematic workflow of how genetic risk genes were identified using H-MAGMA in Supplemental Figure 2 (below), and has been indicated in the text when H-MAGMA is first mentioned in the results section (lines 189-191):

“H-MAGMA was used to identify disease-risk genes in microglia, neurons and oligodendrocytes for AD, PD, MS, ALS and schizophrenia by incorporating PLAC-seq interactomes for the corresponding cell types (Supplemental Fig. 2).”

Response: Thank you for adding this figure. However, the major aspects of the methods that are difficult to follow are the different interaction data that were considered. Therefore, a figure visualizing each of the different interactions would be more beneficial.

Reviewer #3: The authors have responded thoroughly and thoughtfully to all critiques, making eloquent and substantive revisions. The manuscript reads well and effectively addresses prior concerns. I find the revised version to be of high quality and ready for publication.

**Have all data underlying the figures and results presented in the manuscript been provided?**

Reviewer #1: Yes

Reviewer #3: Yes

PLOS authors have the option to publish the peer review history of their article (what does this mean? ). If published, this will include your full peer review and any attached files.

**Do you want your identity to be public for this peer review?** For information about this choice, including consent withdrawal, please see our Privacy Policy .

Reviewer #1: No

Reviewer #3: No

**Figure resubmission:**
---

## [Editor Report · Decision Letter 2]

24 Mar 2025

Dear Dr Nott,

We are pleased to inform you that your manuscript entitled "Genetic risk for neurodegenerative conditions is linked to disease-specific microglial pathways" has been editorially accepted for publication in PLOS Genetics. Congratulations!

Yours sincerely,

Annie Vogel Ciernia

Guest Editor

PLOS Genetics

Wendy Bickmore

Section Editor

PLOS Genetics

Aimée Dudley

Editor-in-Chief

PLOS Genetics

Anne Goriely

Editor-in-Chief

PLOS Genetics

Comments from the reviewers (if applicable):

**Data Deposition**

http://datadryad.org/submit?journalID=pgenetics&manu=PGENETICS-D-24-00990R2

**Press Queries**

---

## [Editor Report · Acceptance letter]

PGENETICS-D-24-00990R2

Genetic risk for neurodegenerative conditions is linked to disease-specific microglial pathways

Dear Dr Nott,

We are pleased to inform you that your manuscript entitled "Genetic risk for neurodegenerative conditions is linked to disease-specific microglial pathways" has been formally accepted for publication in PLOS Genetics! Your manuscript is now with our production department and you will be notified of the publication date in due course.

With kind regards,

Anita Estes

PLOS Genetics

On behalf of:
